# The Patent Ductus Arteriosus in Extremely Preterm Neonates Is More than a Hemodynamic Challenge: New Molecular Insights

**DOI:** 10.3390/biom12091179

**Published:** 2022-08-25

**Authors:** Anna Sellmer, Tine Brink Henriksen, Johan Palmfeldt, Bodil Hammer Bech, Julie Astono, Tue Bjerg Bennike, Vibeke Elisabeth Hjortdal

**Affiliations:** 1Department of Pediatrics, Aarhus University Hospital, Palle Juul-Jensens Boulevard 99, 8200 Aarhus, Denmark; 2Perinatal Epidemiology Research Unit, Department of Pediatrics, Aarhus University Hospital, Palle Juul-Jensens Boulevard 99, 8200 Aarhus, Denmark; 3Department of Cardiothoracic Surgery, Rigshospitalet, Blegdamsvej 9, 2100 Copenhagen, Denmark; 4Research Unit for Molecular Medicine, Department of Clinical Medicine, Aarhus University Hospital, Palle Juul-Jensens Boulevard 99, 8200 Aarhus, Denmark; 5Department of Public Health, Aarhus University, Bartholins Allé 2, 8000 Aarhus, Denmark; 6Department of Health Science and Technology, Aalborg University, 9220 Aalborg, Denmark

**Keywords:** proteomics, patent ductus arteriosus, inflammation, periostin, preterm neonate, angiotensinogen

## Abstract

Complications to preterm birth are numerous, including the presence of a patent ductus arteriosus (PDA). The biological understanding of the PDA is sparse and treatment remains controversial. Herein, we speculate whether the PDA is more than a cardiovascular imbalance, and may be a marker in response to immature core molecular and physiological processes driven by biological systems, such as inflammation. To achieve a new biological understanding of the PDA, we performed echocardiography and collected plasma samples on day 3 of life in 53 consecutively born neonates with a gestational age at birth below 28 completed weeks. The proteome of these samples was analyzed by mass spectrometry (nanoLC-MS/MS) and immunoassay of 17 cytokines and chemokines. We found differences in 21 proteins and 8 cytokines between neonates with a large PDA (>1.5 mm) compared to neonates without a PDA. Amongst others, we found increased levels of angiotensinogen, periostin, pro-inflammatory associations, including interleukin (IL)-1β and IL-8, and anti-inflammatory associations, including IL-1RA and IL-10. Levels of complement factors C8 and carboxypeptidases were decreased. Our findings associate the PDA with the renin-angiotensin-aldosterone system and immune- and complement systems, indicating that PDA goes beyond the persistence of a fetal circulatory connection of the great vessels.

## 1. Introduction

Preterm birth is the leading cause of neonatal death and there is a considerable risk of lifelong impairment [1]. A patent ductus arteriosus (PDA) is a frequent complication to preterm birth. Much emphasis has been placed on the hemodynamic effects of the PDA. However, preterm birth is associated with complex, mutually overlapping abnormalities resulting from systemic immaturity. Hence, the risk attributable to the PDA is most likely not only dependent on gestational age (GA) and the hemodynamics effects of the PDA, but also on various concomitant perinatal factors, including fetal growth restriction, hypoxia, infection, and possibly inflammation [2,3,4,5,6].

Mounting evidence points to a negative impact of early, sustained or intermittent inflammatory exposure on the outcome of preterm birth [4]. A major concern is the possible association between inflammation and immune dysfunction and abnormal stress responses, not only in the immediate postnatal period, but also possibly throughout life. Inflammation is thought to have a direct effect on the function and further development of organs with long-term compromised function of the central nervous system (CNS), as well as respiratory and cardiovascular systems. Despite abundant evidence of these associations, there are significant gaps in our biological understanding of these associations [4,7]. 

Complications to prematurity are complex diseases and their pathogenesis depends on the interaction of a susceptible host with a multitude of environmental and possibly genetic risk factors mutually percolating in a systemic manner, which can be reflected in the plasma protein pattern. Proteins are effectors of biological functions and their levels are not only dependent on corresponding mRNA levels, but also on post translational control and regulation [8]. Therefore, the proteome seems highly relevant to characterize a biological system [9]. Proteomics hold great potential for new insights into pathogenesis related to the continuing presence of a ductus arteriosus and possible novel biomarkers for diagnostic, prognostic, and monitoring purposes in preterm neonates [10]. 

We aimed to investigate proteomics analysis and various cytokines and chemokines in plasma samples from 53 neonates born at a gestational age below 28 weeks to achieve a new biological understanding of the PDA.

## 2. Methods

### 2.1. Study Cohort 

From 1 June 2010 to 28 February 2012, all neonates born with a GA below 32 completed weeks that were admitted to our level three neonatal intensive care unit (NICU), were eligible for inclusion into the PDA study (n = 184). Neonates with chromosomal abnormalities or congenital heart malformations other than atrial septum defects were excluded. In the present study, we included neonates born with a GA below 28 completed weeks, collected as part of the original and previously presented cohort [11]. Parents provided their informed consent for inclusion of their child in the study. The study was conducted in accordance with the Declaration of Helsinki, and approved by the Central Denmark Region Committees on Health Research Ethics (journal number M-20090243) in January 2010, the Danish Data Protection Agency, and the National Board of Health.

### 2.2. Echocardiography and Clinical Information

According to the local guidelines, all neonates born with a GA below 32 weeks had an echocardiography on day 3 after birth (48–72 h) to evaluate the presence of a PDA and the structure of the heart.

The echocardiography was performed using two-dimensional, B-mode, and color Doppler in standard neonatal windows [12]. A PDA was defined as present if flow could be visualized by color Doppler. The PDA diameter was measured in B-mode at the most narrow point. In addition, the PDA was defined as large if the diameter was above 1.5 mm and small if the diameter was 1.5 mm or below [11]. The ratio of the left atrium to the aorta (LA:Ao ratio) was determined in the parasternal long axis view by M-mode using the leading edge method. A large LA:Ao ratio was defined as a ratio above 1.5. Descending aorta diastolic flow (DADF) was evaluated by Doppler flow patterns in the descending aorta obtained from the suprasternal view. None of the neonates in the present study received Ibuprofen or other medicines to close the PDA and none had surgical closure before sample collection.

The following clinical information was obtained from the patients’ medical records: Maternal preeclampsia, antenatal steroid administration, mode of delivery, multiple birth, birth weight, and GA at birth (based on early ultrasound scanning), Apgar score at 1 and 5 min, surfactant administration, intraventricular hemorrhage (IVH) (graded according to Papile et al. [13]), packed red blood cell transfusion within the first 3 days of life, inotropes within the first 3 days of life, early onset sepsis (EOS) defined as 7 days of antibiotics initiated before day 3, and the use of mechanical ventilation on day 3. 

### 2.3. Plasma Samples

Blood samples for this study were obtained along with routine samples on day 3 after birth. The blood was collected by EDTA tubes and centrifuged. The plasma fraction was collected and stored at −80 °C until analysis [14]. N-terminal pro-natriuretic B-type peptide (NT-proBNP) was measured as a routine analysis at the Department of Clinical Biochemistry, Aarhus University Hospital, Denmark [14]. 

### 2.4. Preparation of Samples for Proteomics

To avoid masking of potential biomarkers by the high abundant proteins, the samples were depleted for albumin and IgG. Five μL of plasma was mixed with 10 μL of Complete Mini EDTA-free protease inhibitor. The 53 samples were depleted using ProteoPrep^®^ Immunoaffinity Albumin and IgG Depletion columns. The protein concentration was determined using the Bradford assay (Bio-Rad, Hercules, CA, USA) using technical duplicates, according to the manufacturer’s instructions. Amounts corresponding to 15 μg protein were transferred to a new Eppendorf tube and proteins were precipitated overnight with six times sample volume of acetone at −20 °C, then centrifuged at 2600× *g*, 4 °C for 10 min, and finally air dried. Before mass spectrometry (MS) analysis, the samples were solubilized in 30 µL ammonium bicarbonate with 3% sodium deoxycholate, then reduced with 1.4-dithiothreitol at a final concentration of 5 mM, alkylated in the dark for 30 min with iodoacetamide at a final concentration of 10 mM, and digested with 0.3 μg trypsin. The peptides were subsequently purified with PepClean™ C18 Spin Columns (Thermo Scientific, Waltham, WA, USA), according to the manufacturer’s instructions and then evaporated on a miVac Duo Concentrator (Genevac, Ipswich, United Kingdom). Samples were stored at −20 °C.

### 2.5. LC-MS/MS Analysis 

The peptides were re-suspended in 2% ACN, 0.1% formic acid, and were analyzed by nanoLC-MS/MS, essentially as described previously [15]. The samples were analyzed by nano-liquid chromatography (Easy-nLC 1200, Thermo Scientific)-tandem MS (Q-Exactive HF-X Hybrid Quadrupole Orbitrap, Thermo Scientific). Peptides were trapped by a pre-column (Acclaim PepMap 100 C18, pore size: 100 Å, particle diameter: 3 µm, inner diameter: 75 µm, length: 2 cm, Thermo Scientific) and separated further with a reverse phase analytical column (PepMap RSLC C18, pore size: 100 Å, particle diameter: 2 µm, inner diameter: 75 µm, length: 25 cm, Thermo Scientific) using a 100 min gradient from 5–90% ACN and 0.1 formic acid at a 270 nL/min flowrate. The mass spectrometer was operated in positive mode and higher collision dissociation (HCD) at normalized collision energy of 29 was used for peptide fragmentation. The full scan/MS1 resolution was 60,000, AGC target was 3 × 10^6^, maximum injection time was 80 ms, and scan range was from 340 to 1700 *m*/*z*. The fragmentation scan/MS2 resolution was 15,000 and the automatic gain control (AGC) target was 1 × 10^5^. The MS was operated in data dependent mode and up to 10 of the most intense peaks were fragmented. Dynamic exclusion was set to 30 s and both unassigned and single charge ions were excluded. 

### 2.6. Proteomic Data Analysis 

Proteins were identified and quantified using MaxQuant (version 1.5.3.30, https://www.maxquant.org/, accessed on 5 July 2022) with the building Andromeda algorithm against the human sequence database (Homo Sapiens proteome with 20,129 reviewed sequences from Uniprot.org, accessed on 5 July 2016). Settings included the enzyme trypsin with a maximum of two missed cleavage sites; precursor mass tolerance: 10 ppm; fragment mass tolerance: 0.02 Da; dynamic modification: Oxidation; static modification: carbamidomethyl; and FDR was 0.01 at protein and peptide level. To maximize identifications in MaxQuant, files containing MS spectra from 10 fractions of a pooled study of plasma samples, were uploaded along with MS data from the samples and the “match between runs” were applied.

### 2.7. Multiplexed Luminex Analyses

A total of 19 proteins with relation to inflammation were simultaneously quantitated by the bead-, antibody-, and fluorescence-based method Luminex. The 19-plex arrays, including VEGF-A, GM-CSF, TNF- α, IL-1ra, IL-1 alpha, IL-1 beta, IL-4, IL-6, IL-8 (CXCL8), IL-10, IL-12p70, Eotaxin (CCL11), SDF-1 alfa, CD62E (E-selectin), MCP-1 (CCL2), MIP-1, lfa (CCL3), MIP-1 beta (CCL4), RANTES (CCL5), and ICAM-1 were analyzed on the Luminex XMAP platform, Magpix (Millipore Corp, MA, USA), according to the manufacturer’s instructions. 

### 2.8. Statistical and Bioinformatics Analyses

The data analysis was performed in R v4.1.2 (R Core Team), using the LFQ-columns from the MaxQuant generated proteingroups.txt file, which contains protein identifications filtered to the false discovery rate (FDR) < 1%.

Additional filtering was performed to ensure high accuracy qualitative data, by removing (i) proteins tagged as reverse, contaminants, or only identified by modified peptides; (ii) proteins where the quantitation was based on less than one unique peptide to a given protein group; and (iii) proteins that were quantifiable in less than 70% of samples in both of the compared groups, i.e., large PDA compared to no PDA or small PDA compared to no PDA, thereby ensuring that at least 70% of samples in both groups were compared. The remaining missing values were not included in the differential analysis. To include condition-unique proteins, which would have been filtered using the applied filtering strategy, we performed a Fisher’s Exact analysis for all proteins, but none passed the statistical filtering criteria (*p*-value < 0.05). Solely for the purpose of conducting principal component analysis (PCA), missing data were imputed by random draws from a Gaussian distribution centered below the minimal value observed in that sample (q = 0.01, tune.sigma = 0.3) to simulate signals from low-abundant proteins [16].

In the days after term birth, the plasma concentration of several proteins is known to change, since especially the innate immune system, it rapidly develops in response to new environmental exposures [17]. Therefore, we used linear mixed-effects regression models to identify PDA-associate proteins. A full linear mixed-effects regression model of LFQ values was fitted with fixed effects of PDA status and a random GA week effect, using the lmer function from the lme4 R package [18]. The *p*-values were calculated using the ANOVA function against a null model, omitting the PDA status, but including GA to take the ontogeny into account. The *p*-values were corrected for multiple hypothesis testing using the Benjamini–Hochberg approach [19]. Proteins were considered statistically significantly different at adjusted *p*-values < 0.05 and +/− 0.3 log2 fold-change. To expand the analysis, we included an analysis of all proteins at unadjusted *p*-value < 0.05. The analysis was performed in R v4.1.2 [20] using Rstudio (2021.09.0) [21]. The packages dplyr [22] and mixOmics [23] were used for data formatting, and plots were performed using ggplot2 [24], ggpubr [25], and cowplot [26]. Significantly differentiating proteins were submitted to String-DB to infer known protein–protein interactions [27]. Receiver operating characteristic (ROC) curves were generated using the pROC package, and *p*-values were calculated using the test.roc function [28].

### 2.9. Periostin Enzyme-Linked Immunosorbent Assay

To validate the MS analysis, an enzyme-linked immunosorbent assay (ELISA) targeting periostin was performed using the Human Periostin ELISA Kit (Thermo Fisher). Periostin was chosen as a protein target for the following reasons: (i) It is currently suggested as a biomarker in preterm neonates with BPD [29,30] and (ii) it was one of the proteins found at higher levels in neonates with PDA by LC-MS proteomics. The test was carried out following the manufacturer’s instructions with a minor modification. The instructions suggest that a 1:2 dilution of plasma samples deviations was performed as a preliminary test of the kit and another study of plasma periostin in connection with BPD [29] indicated that a dilution of 1:33 would be more suitable for the absorbance not to exceed the measurable absorbance of the plate reader. Therefore, samples were diluted 1:33 with the sample diluent provided in the kit. The absorbance was measured at 450 nm using a synergy H1 microplate reader (BioTek, Winooski, VT, USA). A standard curve was based on the periostin standard solutions and concentrations in the samples were determined.

## 3. Results

### 3.1. Study Cohort

The study cohort comprised 53 newborns, 33 (62%) were determined to have a PDA (Table 1) on day 3 of life. We have previously demonstrated that a PDA with a diameter above 1.5 mm at this day in extremely preterm neonates is associated with adverse outcomes [11]. Accordingly, we divided our cohort into the following three groups: No PDA (n = 20), small (n = 13), and large PDA (n = 20). Neonates with a large PDA had more unfavorable baseline characteristics compared to neonates with no PDA; however, they were not statistically significant. 

### 3.2. Plasma Proteomics

LC-MS-based proteomics enabled us to monitor the relative abundance of 219 proteins which passed our stringent filtering, ensuring high-confidence identifications and quantitation. Furthermore, 18 cytokines and chemokines monitored by immunoassays passed the valid value-filtering criteria. To investigate the global proteomics data, we performed an unsupervised principal component analysis (PCA). No clear outlying samples could be identified on the PCA scores plot of the LC-MS proteomics data (Figure 1A) nor immunoassay data (Figure 1B). Moreover, no clear grouping of the samples could be identified based on the available clinical parameters, including GA or PDA. However, the overlap of the no PDA and small PDA samples was generally larger than the large PDA samples, indicating a higher degree of similarity.

### 3.3. Identifying Proteins Differentiating PDA from No PDA Samples

To further expand our understanding of the biological mechanisms of PDA, we performed a differential analysis to identify proteins with a PDA-specific concentration profile.

Comparing the small PDA to no PDA, no proteins passed the statistical filtering criteria described in the Methods section, including the adjusted *p*-value < 0.05 (Appendix A) (full list of proteins and cytokines in Appendix A, respectively).

Comparing the large PDA samples to no PDA samples, two proteins and three cytokines passed our statistical correction for multiple hypothesis testing, including the adjusted *p*-value < 0.05 (Figure 2A,B for LC-MS proteomics and immunoassay, respectively) (full list of proteins and cytokines in Appendix A, respectively).

To investigate if the lack of significant proteins comparing small PDA to no PDA reflected the smaller sample size compared to large PDA vs no PDA, we compared the protein fold-changes between small PDA vs no PDA to large PDA vs no PDA. A statistically significant correlation was found for both LC-MS (R_spearman_ = 0.57, *p*-value = 2.2 × 10^−16^) and immunoassay proteins (R_spearman_ = 0.66, *p*-value = 3.7 × 10^−3^). However, the protein fold-changes were generally smaller for the small PDA as demonstrated by the < 1x slope of a linear regression for LC-MS (y = 0.47x − 0.02, Appendix A) and immunoassay proteins (y = 0.41x + 0.14, Appendix A) suggesting a dose-response-like relationship between PDA size and protein levels.

Therefore, we focused our study on the comparison between the large PDA and no PDA. For hypotheses generation, we included the additional 19 proteins and 5 cytokines with different abundances between large PDA and no PDA, which passed our statistical filtering criteria, but with the unadjusted *p*-value < 0.05 in the analysis (Table 2 and Table 3, respectively). Of the LC-MS proteins, 15 were increased and 6 decreased in large PDA compared to no PDA, whereas all investigated cytokines were increased. The proteins included, but were not limited to, angiotensinogen (AGT), osteopontin (SPP1), periostin (POSTN), and the cytokines, included IL-6, IL-8, IL-10, and IL-1RA. The decreasing proteins were mainly related to the complement system, e.g., C8 (alpha, beta, and gamma chain) and carboxypeptidase N catalytic chain CPN1 and CPN2.

To investigate whether the significant proteins were associated with PDA rather than ontogeny, we repeated all of the calculations using only samples from the GA 27 group (Appendix A). A positive correlation was found between the large PDA compared to no PDA protein fold-changes (R_spearman_ = 0.75, *p*-value = 1.42 × 10^−4^). Additionally, we plotted the protein profiles for all significant large PDA compared to no PDA proteins (Appendix A).

A joint protein–protein interaction analysis of all proteins and cytokines, revealed that known interactions between the proteins and cytokines/chemokines increased and decreased during PDA, respectively, with little overlap (Figure 2C). The finding indicates the presence of PDA interactions between the proteins and cytokines, distinct functional changes in PDA, and the increased involvement of the immune system in neonates with PDA.

### 3.4. Immunoassay Validation of Periostin (POSTN) Levels

To ensure the consistency in protein measurements across methods, we compared the relative levels of plasma periostin (POSTN) as determined by the fundamentally different protein measurement techniques LC-MS/MS and immunoassay. A strong and significant sample–sample correlation was determined (Figure 3A) (R_spearman_ = 0.73, *p*-value = 9.4 × 10^−10^).

### 3.5. PDA and Increased Levels of NT-proBNP Levels

We re-analyzed the plasma levels of the known PDA-marker NT-proBNP in part of the original cohort [14], which was included in the present study. The marker was found to be significantly higher in neonates with PDA (Figure 3B) (mean PDA level = 389%, *p*-value = 1.67 × 10^−8^, *q*-value = 6.68 × 10^−7^), as expected for a PDA cohort. Moreover, we generated receiver operating characteristic (ROC) curves (Figure 3C), which demonstrated that NT-proBNP can separate large PDA from no PDA with an accuracy of 93.4%. In comparison, the AUC was 60.1% for the ROC compared with the small PDA compared to no PDA.

## 4. Discussion

### 4.1. Short Presentation of Main Results

In a cohort of 53 extremely preterm neonates, we found that proteomics analysis and multiplex ELISA revealed evidence of protein abundance differences in plasma related to multiple biological immune system associated processes, including coagulation, complement activation, inflammation, and immunomodulation. The two proteins that mainly differed between extremely preterm neonates with large PDA compared to no PDA were angiotensinogen (AGT) with a 1.5-fold increase and periostin with a 1.7-fold increase in the LC-MS analysis. Moreover, IL-1RA, IL-6, IL-8, and IL-10 were found at higher levels in neonates with large PDA in the ELISA analysis. Contrary lower levels of complement factors C8 and carboxypeptidases were found in neonates with large PDA compared to no PDA.

The PDA is a dynamic structure and the small PDA may even close from time to time. We have previously demonstrated that a large PDA in extremely preterm neonates is associated with adverse outcomes [11]. With the present study, we demonstrate that although the general direction of the proteome fold-changes was identical, we found overall more pronounced protein fold differences in neonates with large PDA vs. no PDA, compared to small PDA vs. no PDA.

### 4.2. Activation of the RAAS System in Neonates with PDA

Angiotensinogen (AGT), the precursor of all angiotensin peptides, was found in higher levels (1.5 fold-change) in neonates with large PDA compared to neonates with no PDA. AGT is a member of the serpin superfamily, as well as alpha1 antitrypsin, which is also found at higher levels in neonates with large PDA.

In circulation, AGT is cleaved to angiotensin-I by renin, which is secreted from the juxtaglomerular apparatus in the kidneys in response to a decreased renal perfusion. However, also immune mediators, such as IL-6, are involved [31,32]. We found IL-6 at higher levels in neonates with large PDA compared to neonates with no PDA. Angiotensin I is subsequently converted to angiotensin II (Ang II) by the angiotensin-converting enzyme (ACE).

Ang II is recognized not only as a physiological mediator restoring circulatory integrity, but also as a growth factor that regulates cell growth and fibrosis, organ differentiation, and a key element in the inflammatory process [33,34]. Ang II increases the vascular permeability that initiates the inflammatory process [35] and contributes to the recruitment of inflammatory cells [33]. Moreover, there is an increasing evidence that pro-inflammatory factors enhance the expression of RAAS components [36].

### 4.3. Ang II and IL-10 Stimulated Increase in SPP1 in Neonates with PDA

SPP1 was the protein with the second largest fold-change in neonates with large PDA compared to neonates with no PDA in the LC-MS data. Previously, an association between cord blood SPP1 and PDA has been demonstrated [37], but this is the first study to demonstrate higher levels in plasma samples from neonates with large PDA.

Secreted phosphoprotein 1 (SPP1), also known as osteopontin (OPN), is a matricellular protein that mediates diverse biological functions. SPP1 functions as a pro-inflammatory cytokine and promotes cell-mediated immune responses [38]. In addition, it has protective functions, such as biomineralization and wound healing. During pathologic processes, SPP1 is produced by various cells [39,40].

Rat studies have demonstrated that Ang II can induce SPP1 expression. And further that the inhibition of the Ang II type 1 receptor (AT1) attenuate the expression of SPP1 [41,42,43]. Moreover, SPP1 expression is stimulated by inflammatory cytokines, including IL-10 [44]. We found higher levels of IL-10 in neonates with large PDA compared to neonates with no PDA. It is thought that the acute increase in SPP1 has a protective role in cardiovascular disease in adults, whereas a more chronic increase predicts poor prognosis [45]. The function and regulation of SPP1 in preterm neonates remain elusive [45,46].

### 4.4. Which Cytokines Are Associated with Large PDA?

Cytokines are pleiotropic endogenous inflammatory and immunomodulating mediators that exhibit regulatory effects on various target cells [47]. These cell-derived polypeptides are involved in both acute and chronic inflammatory processes by acting locally or systemically. The IL-1RA was the interleukin with the highest difference in concentration between neonates with large PDA and no PDA amongst the measured cytokines. IL-1RA is a naturally occurring cytokine that inhibits the effects of IL-1α and IL-1β through competitively ligand-specific binding to IL-1 receptors without exhibiting detectable agonist activity [48]. We found higher IL-1β levels in neonates with a large PDA compared to neonates with no PDA. IL-1α and IL-1β are potent early response inflammatory cytokines that modulate their own production and induce other pro-inflammatory cytokines, including IL-6, IL-8, and tumor necrosis factor (TNF)-alpha. The biological effects of IL-1 range from inducing specific cell responses to targeting entire systems and may be both important for host responses to injury and infection and pathological in other conditions [49].

The underlying cause of the association between pro-inflammatory markers and large PDA in very preterm neonates is unknown. Speculatively, in small-scale clinical studies, an alteration in inflammatory response has been found at baseline in children with congenital heart disease (CHD) [50,51]. Moreover, an immune contribution to the CHD development has been suggested [52], as well as compensatory remodeling mechanism caused by changes in the cytokine profile. Children born with an atrial septal defect (ASD) undergo cardiac remodeling to compensate for the flow of blood from the left to the right atrium. In these children, the presence of markers for mechanical stress, inflammation, and remodeling are noted [53]. Similar differences in acute-phase reactants are found in relation to ventricular septal defects (VSDs) [54].

### 4.5. Balancing Pro- and Anti-Inflammation in the Preterm Neonates

We found that neonates with large PDA had higher plasma levels of IL-6, IL-8, IL1RA, and IL-10 compared to neonates with no PDA. This is supported by previous findings, in a study including 47 neonates born at a GA below 28 weeks, which revealed that IL-6, IL-8, IL-10, IL-12 and growth differentiation factor 15 and monocyte chemotactic protein 1, were associated with hemodynamically significant PDA. This was defined as a PDA with ductal diameter of >1.5 mm, or a left atrium to aorta ratio of >1.5, or absent or reversed flow during diastole in the descending aorta [55].

A rapid systemic inflammatory response can be an effective defense against microbial invasion. However, failure of inflammation-resolution processes leads to dysregulated and prolonged inflammation, even possibly systemic inflammation [56]. Several studies indicate that cytokine dysregulation during the first week of life, most notably high levels of IL-8, is associated with long-term morbidity, including BPD [57], NEC [58], atypical white matter brain development, and executive function limitations in adolescence [59,60].

We found higher anti-inflammatory IL-10 and IL-1RA levels in neonates with large PDA. This may not be sufficient to balance the pro-inflammatory response present in neonates with large PDA driven by IL-1beta, IL-6, IL-8, and other inflammatory plasma proteins. The premature infant may be at a particularly high risk for unopposed pro-inflammatory effects, since developmental immaturity may limit the ability to increase endogenous expression of anti-inflammatory mediators sufficiently [56]. However, other sustained inflammatory stimuli from, e.g., mechanical ventilation, organ damage, and epigenetics processes, that are unable to shift from a pro-inflammatory to an anti-inflammatory state, may also contribute to intermittent or sustained systemic inflammation.

### 4.6. Complement System

The complement system is an essential part of the innate immune response. In addition, it functions as the first line of defense against pathogens and elicits a pro-inflammatory response, leading to recruitment and activation of immune cells from both the innate and adaptive branches of the immune system [61]. In neonates with large PDA, we found complement factor 8 (C8) alpha, beta, and gamma chain at lower plasma levels. A study including 60 infants (average, 4 years) with PDA and no other congenital heart disease and 60 controls with no PDA found that plasma levels of C3, C7, C8, and C9 were lower in infants with PDA compared to infants with no PDA [62].

Furthermore, C3 was decreased in the present study in neonates with large PDA, but with a significantly small fold-change to pass our filtering. Moreover, infants with congenital heart defects have decreased serum levels of C3 and C4 compared to infants without structural heart disease [63]. The biological properties of C3a are regulated by the protein carboxypeptidase N (CPN). In neonates with large PDAs, we found lower plasma levels of carboxypeptidase N catalytic chain (CPN) 1 and carboxypeptidase N subunit 2 (CPN2). Complement components are synthesized early in fetal life, but with a relative deficiency in comparison with adult levels [64]. It is unclear if the low complement levels in neonates with PDA are related to issues with the liver, the main synthesis location for most complement components, or due to the activation and consumption of complement factors.

### 4.7. Is Periostin a Marker of Pressure Overload or of Pulmonary Remodeling?

The protein with the highest fold-change concentration in neonates with large PDA compared to neonates with no PDA was periostin. To the best of our knowledge, this is the first study to demonstrate that extremely preterm neonates with large PDA have higher plasma levels of periostin compared to neonates with no PDA. Periostin is a matricellular protein with functions in osteology, tissue repair, oncology, cardiovascular and respiratory systems, and in various inflammatory settings and diseases [65].

The hemodynamic effects of a large left-to-right shunt associated with a PDA includes hyperperfusion of the pulmonary vasculature and volume overload of the atrium and ventricles. In the adult heart, periostin is almost undetectable [66], but is induced in the ventricles following myocardial infarction, pressure overload, or generalized cardiomyopathy [67]. Speculatively, the hemodynamic effects of the large PDA may, at least in part, explain the higher levels of periostin. Supporting this, NT-proBNP is also known to be increased in relation to circulatory volume overload, and along with other authors, we have demonstrated that NT-proBNP is increased in neonates with PDA compared to neonates with no PDA [14]. Data on NT-proBNP were re-analyzed in this cohort, found to be increased, and as a single marker, were able to separate large PDA from no PDA with an outstanding discrimination. We speculate that a large PDA induces the expression and release of periostin and NT-proBNP from the heart due to volume overload.

Periostin levels may represent pulmonary pathology as periostin expression is increased in lungs of neonates that died with severe BPD and is currently evaluated as a marker of BPD [30]. Periostin has been reported to be associated with neonatal murine lung remodeling [68] and hyperoxic lung injury [69].

In adults, periostin expression in the lungs decreases following acute injury, but then increases substantially during the initiation of repair-mechanisms and even beyond the initial insult, this increase may persist [65]. Therefore, periostin may in the context of a PDA be an important structural mediator, conveying tissue adaption in response to insult or injury [65,70].

### 4.8. Do the Levels of sCD163 Indicate Inflammation Involving Macrophages System?

On proteomic analysis, we found higher levels of sCD163 in neonates with large PDA compared to neonates with no PDA. The scavenger receptor CD163 is expressed in macrophages and monocytes and is a receptor for multiple ligands, which are quantitatively important for the haptoglobin-hemoglobin complex [71]. Increased plasma concentration of soluble CD163 (sCD163) [72] is observed in diseases related to macrophage activity, including acute and chronic inflammations in adults [71].

In vitro, CD163 is upregulated by glucocorticoids, hemoglobin, and both IL-10 and IL-6 [73]. We speculate that higher sCD163 levels, also supported by higher levels of IL-6 and IL10 in neonates with large PDA, reflect sustained inflammation, or are also related to PRBC transfusion in neonates with large PDA.

### 4.9. Strengths and Limitations

In this study, the cohort investigated was comprised of 53 extremely preterm neonates. Very few studies have reported both clinical information and plasma proteomics data from LC-MS and ELISA analysis from this unique patient group. Our advanced laboratory and statistical methods allowed us to assess both ontogeny and the effect of GA when we identified proteins and inflammatory markers with a statistically significantly PDA associated abundance. Blood was not drawn at the exact same postnatal age (by hours). However, the postnatal ontogenetic effects by age on the plasma proteome is assumed to be minimal compared to prenatal, i.e., taking gestational age into account was crucial. Some heterogenicity may have caused a low statistical power and we included proteins that do not pass multiple hypothesis test corrections in our interpretations. However, many of the included proteins were functionally highly similar in supporting this decision.

Preterm neonates are a heterogenic patient population not only by GA at birth, but also by reason for their preterm birth and their postnatal challenges. We aimed to describe characteristics of the infants present prior to blood sampling and echocardiography postnatal day 3. Preeclampsia, chorioamnionitis, and use of antenatal steroids could impact the risk of both PDA and also influence the proteome. Moreover, both RDS (hence use of surfactant) and sepsis could contribute to an inflammatory response. Unfortunately, we do not have data on chorioamnionitis. Neither preeclampsia, RDS, nor sepsis was found more often in neonates with large PDA compared to neonates with no PDA in this cohort. However, mechanical ventilation was more frequently used in neonates with large PDA and might be an external stimuli for inflammation. What changes PRBC transfusion pose on the proteome has not been fully described.We found coherence between the results obtained with LC-MS and ELISA methods of periostin indicative of a technically robust analysis. Nonetheless, the findings should be validated in an independent study, ideally with a narrower GA distribution and larger sample size. Future studies characterizing heart tissue from newborns with PDA by transcriptome in combination with proteomics will likely enable a mapping of the tissue origins of many of the detected proteins [74], and provide further insights into the PDA etiology.

## 5. Conclusions

The present study investigated the proteome, which fluctuates over time in response to internal and external stimuli, making it particularly valuable for understanding various complications in preterm neonates. We found that a large PDA in extremely preterm neonates was associated with differences in angiotensinogen, periostin, and measures of both immune- and complement systems. A small PDA was associated with more subtle biological differences in the plasma proteome. Moreover, our findings indicate that the PDA may interfere with or be driven by an imbalance in core bio-physiological systems. Therefore, the bio-physiological understanding of mechanisms behind inflammatory stimuli and the neonate response to inflammation should be the focus of future PDA research. In particular, it may increase our understanding of pathogenesis, reveal potential tools for early diagnosis, drug development and prognostication, and present ways to monitor disease severity or improvement.

## Figures and Tables

**Figure 1 biomolecules-12-01179-f001:**
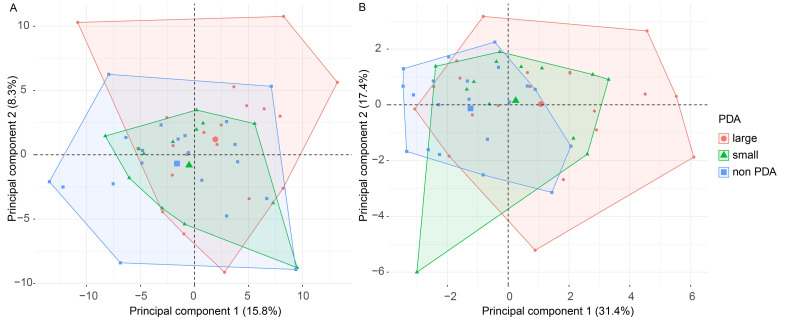
**Unsupervised principal component analysis (PCA)**. Unsupervised PCA scores plot of the (**A**) LC-MS proteomics data and (**B**) immuno-based assay data. Each dot represents a sample and the color denotes the PDA size. Explained variance is provided in percentages. No clear outliers were identified.

**Figure 2 biomolecules-12-01179-f002:**
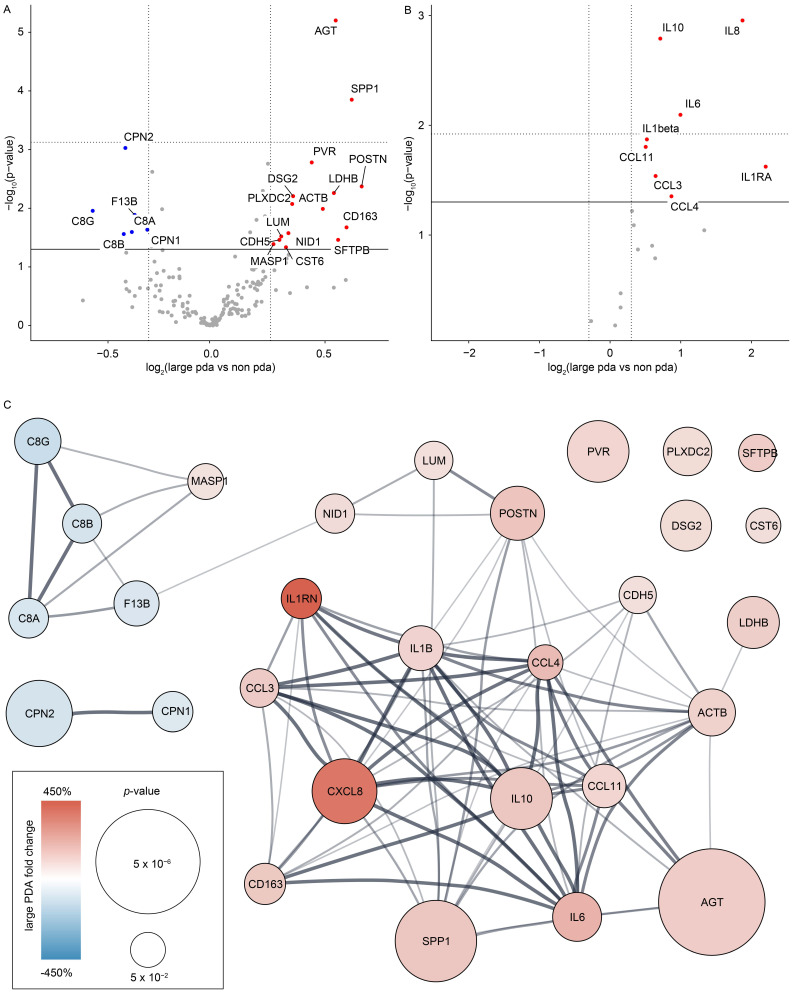
**Large PDA and the plasma proteome.** Data from 53 neonates born at a gestational age below 28 weeks. (**A**) Plasma proteome changes large PDA compared to no PDA as determined by LC-MS proteomics, and (**B**) immunoassay. Red (blue) proteins are increased (decreased) in PDA. Grey proteins are not significantly different. Gene names are provided. Horizontal solid line: *p*-value = 0.05, horizontal dotted line: *q*-value = 0.05, vertical dotted lines: Fold-change cutoff. (**C**) Combined analysis of the large PDA compared to no PDA changing plasma proteins. Color and protein size indicates PDA fold-change and statistical significance, respectively. Known protein–protein interactions from the STRING database are indicated by lines, where the width signifies the number of known interactions.

**Figure 3 biomolecules-12-01179-f003:**
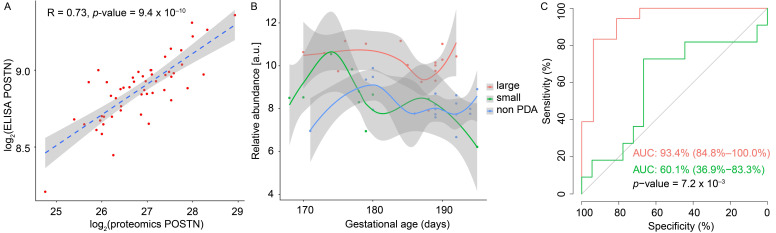
**LC-MS proteomics and immunoassay data validation.** (**A**) Correlation of periostin (POSTN) plasma levels in all samples as determined by LC-MS and immunoassay. Pearson’s correlation coefficient (R), *p*-value, and 95% confidence interval (grey) is provided. (**B**) NT-proBNP plasma abundance variation over the gestational ages. (**C**) Sensitivity and specificity of NT-proBNP to identify large PDA (red) and small (green) PDA compared to no PDA, respectively. Area under the curve (AUC) and *p*-value for differences between the ROCs are provided.

**Table 1 biomolecules-12-01179-t001:** Study cohort.

	no PDA (n = 20)	Small PDA (n = 13)	Large PDA (n = 20)	*p*-Value *
GA, median (range)	27 (24; 27)	25 (24; 27)	26 (24; 27)	n.s.
Birth weight grams, median (range)	984 (545; 1220)	750 (575; 1000)	902 (570; 1210)	n.s.
Sex (female/male)	6/14	4/9	7/13	n.s.
Apgar score at 1 min (IQR)	7 (3; 9)	6 (4; 9)	7 (5; 8)	n.s.
Apgar score at 5 min (IQR)	10 (8; 10)	10 (8; 10)	10 (8; 10)	n.s.
Caffeine, number (%)	20 (100)	13 (100)	20 (100)	n.s.
EOS, number (%)	4 (20)	6 (46)	4 (20)	n.s.
Surfactant, number (%)	9 (45)	7 (54)	13 (65)	n.s.
PRBC transfusion, number (%)	4 (20)	5 (38)	7 (35)	n.s.
Inotropes, number (%)	1 (5)	1 (8)	3 (15)	n.s.
Mechanical ventilation, number (%)	3 (15)	5 (38)	9 (45)	0.08
IVH, number (%)	5 (25)	2 (15)	10 (50)	n.s.
Large LA:Ao ratio, number (%)	1 (5)	1 (8)	6 (30)	0.09
Reversed DADF, number (%)	0 (0)	1 (8)	4 (20)	n.s.
Preeclampsia, number (%)	4 (20)	1 (8)	1 (5)	n.s.
Antenatal steroids, number (%)	20 (100)	13 (100)	17 (85)	n.s.
Multiple pregnancy, number (%)	3 (15)	5 (38)	9 (45)	0.08
Cesarean delivery, number (%)	13 (65)	6 (46)	13 (65)	n.s.

Characteristics of 53 neonates born before 28 + 0 gestational weeks by presence of PDA postnatal day 3. Small: PDA diameter ≤ 1.5 mm, large: PDA diameter ≥ 1.5 mm. * *p*-values listed for large PDA to no PDA comparison. Gestational age (GA) weeks, packed red blood cell transfusion (PRBC) within the first 3 days of life, inotropes within the first 3 days of life, early onset sepsis (EOS) defined as 7 days of antibiotics initiated before day 3. Mechanical ventilation used day 3. Intraventricular hemorrhage (IVH). Large LA:Ao ratio (left atrium to aorta ratio) > 1.5. DADF descending aorta diastolic flow.

**Table 2 biomolecules-12-01179-t002:** **Large PDA significant proteins from LC-MS**. Statistically significant proteins (*p*-value < 0.05) differentiating large PDA from no PDA, as determined by proteomics.

Protein Name	PDA Change [log2]	Large PDA/No PDA Ratio	*p*-Value	Gene Name
Periostin	0.75	168%	4.24 × 10^−3^	POSTN
Osteopontin	0.70	162%	1.41 × 10^−4^	SPP1
Scavenger receptor cysteine-rich type 1 protein M130	0.67	160%	2.12 × 10^−2^	CD163
Pulmonary surfactant-associated protein B	0.63	155%	3.46 × 10^−2^	SFTPB
Angiotensinogen	0.62	154%	6.29 × 10^−6^	AGT
L-lactate dehydrogenase B chain	0.61	153%	5.50 × 10^−3^	LDHB
Actin, cytoplasmic 1	0.56	147%	1.03 × 10^−2^	ACTB
Poliovirus receptor	0.50	142%	1.66 × 10^−3^	PVR
Desmoglein-2	0.41	133%	6.22 × 10^−3^	DSG2
Plexin domain-containing protein 2	0.41	133%	8.45 × 10^−3^	PLXDC2
Nidogen-1	0.39	131%	2.66 × 10^−2^	NID1
Cystatin-B	0.38	130%	4.62 × 10^−2^	CST6
Lumican	0.35	128%	3.02 × 10^−2^	LUM
Cadherin-5	0.34	127%	3.46 × 10^−2^	CDH5
Mannan-binding lectin serine protease 1	0.31	124%	4.09 × 10^−2^	MASP1
Carboxypeptidase N catalytic chain	−0.31	81%	2.32 × 10^−2^	CPN1
Coagulation factor XIII B chain	−0.37	77%	1.30 × 10^−2^	F13B
Complement component C8 alpha chain	−0.38	77%	2.54 × 10^−2^	C8A
Carboxypeptidase N subunit 2	−0.41	75%	9.38 × 10^−4^	CPN2
Complement component C8 beta chain	−0.42	75%	2.75 × 10^−2^	C8B
Complement component C8 gamma chain	−0.58	67%	1.11 × 10^−2^	C8G

**Table 3 biomolecules-12-01179-t003:** **Large PDA significant proteins from immunoassay.** Statistically significant cytokines (*p*-value < 0.05) differentiating large PDA from no PDA, as determined by immunoassay.

Interleukine/Cytokine Name	PDA Change [log2]	Large PDA/No PDA Ratio	*p*-Value	Gene Name
Interleukin-1 receptor antagonist	2.20	460%	2.38 × 10^−2^	IL1RA
Interleukin-8	1.87	367%	1.11 × 10^−3^	IL8
Interleukin-6	1.00	199%	8.03 × 10^−3^	IL6
C-C motif chemokine 4	0.87	183%	4.44 × 10^−2^	CCL4
Interleukin-10	0.71	164%	1.62 × 10^−3^	IL10
C-C motif chemokine 3	0.64	156%	2.90 × 10^−2^	CCL3
Interleukin-1 beta	0.52	143%	1.34 × 10^−2^	IL1beta
Eotaxin	0.50	142%	1.57 × 10^−2^	CCL11

## Data Availability

The data can be made available by contacting the authors.

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
