# Peer review of "The Patent Ductus Arteriosus in Extremely Preterm Neonates Is More than a Hemodynamic Challenge: New Molecular Insights"

_biomolecules, 2022, doi:10.3390/biom12091179_

Round 1

Reviewer 1 Report

The study by Sellmer et al on new molecular insights of PDA makes interesting reading. These studies add information regarding the complex molecular mechanisms relating to PDA. This is a challenging study with rigorous methodology and statistical analysis. However, the authors should make clinical observations relevant to the clinician in the management of these infants. The reviewers have the following suggestions.

The authors have commented on small PDA and large PDA based on ductus size. The authors should comment on whether a duct >1.5mm is associated with hemodynamically significant PDA. Do the authors equate a large duct (>1.5mm) to a hemodynamically significant PDA. Hemodynamic compromise in each of the groups should be in Table 1.

As this is a study and not a review, please delete Figure 1. Normally it is not a good practice to have figures in the introduction unless it is a review. The figure is not adding anything to the manuscript. Also, the reviewer does not see any reference to Figure 1 in the introduction.

Figure 1 – The figure has three parts. 1A – is it the diagram of the fetus or the newborn? 1B – the normal heart. The arrow from 1A to 1B means that the 1B should demonstrate a PDA (with a right to left shunt, if 1A is a fetus) or left to right if 1A is a neonate. Additionally, the figure legend does not explain the three figures in Figure 1. Please delete Figure 1.

Methods

In the clinical information section, how many babies in the study received caffeine within the first three days of life or prior to sample collection for analysis? Please clarify caffeine use in the NICUs. Caffeine could impact protein biomarkers in the blood. We need information in Table 1.

The authors should present more information on the clinical course until discharge. The real question is whether protein biomarkers predict morbidities and mortality at 36 weeks of PMA.

How many of the PDAs closed spontaneously in all the groups. Did any infants enrolled in the study treated medically (medications) or by surgery (ligation)? How do they relate to the proteomic change on Day 3?

The authors should present data on BPD at 36 weeks GA in all the groups. Additionally, other major morbidities such as IVH, NEC, and ROP status at 36 weeks should be presented in the Table.

The ventilation status of infants in all the groups on the day of collection of samples is of additional interest. Mechanical ventilation by itself can induce inflammatory change in premature infants.

Discussion – The reviewers have the following questions regarding lower protein fold differences with small PDA vs no PDA, compared to large PDA vs no PDA –

1.       The infants in the large PDA group were of lower gestational age (Table 1). As far as proteomics is concerned, everyday change in GA will produce a change in plasma proteomics in premature infants. Could the proteomic change be a surrogate marker for gestational age rather than PDA itself?

2.       The authors have to highlight how they made the association between proteomic change and PDA status.

3.       Could PRBC transfusions have contributed to a proteomic change, as more babies in the large PDA group had PRBC transfusions.

4.       Sepsis was higher in the small PDA group; Preeclampsia which could contribute to inflammation was higher in the no PDA group. How do the authors relate the findings to the data in Table 1?

5.       Inflammatory markers were increased in the large PDA groups, the clinical data presented in Table 1 does not support the finding.

The authors have to expand on the limitations of the study. The study has a lot of limitations as mentioned above.  

In the absence of clinical factors related to an infection at birth (such as chorioamnionitis), how do the authors explain changes in inflammatory markers as they relate to PDA?

Despite the limitations of the study, the study demonstrates that ductal patency may be influenced by numerous factors. However, is the proteomic change leading to PDA or large PDA leading to proteomic change? The authors should elaborate on this in the discussion, which would be interesting. Also, making the findings clinically relevant will be of additional interest. Presenting more clinical information will help the readers in making reasonable conclusions about this excellent study.

Conclusions. L482-484. The first sentence that ‘our study discloses….’ This is the summary of the study. This sentence is apt for the beginning of the discussion so that the readers follow the discussion better. The conclusions should include how the authors interpret the results and make conclusions based on the results.

The second statement – The study provides novel insights into molecular insights…… The reviewer does not agree with that statement. Please delete the statement. The study demonstrates an association between proteomics and PDA status. Association is not causation. 

Author Response

The study by Sellmer et al on new molecular insights of PDA makes interesting reading. These studies add information regarding the complex molecular mechanisms relating to PDA. This is a challenging study with rigorous methodology and statistical analysis. However, the authors should make clinical observations relevant to the clinician in the management of these infants. The reviewers have the following suggestions.

We thank reviewer 1 for his interest in this study and his acknowledgements of the strengths of this manuscript. The reviewer has important comments to the manuscript and in the following, we have made our best effort to answer. And we appreciate the improvements this has resulted in.

The authors have commented on small PDA and large PDA based on ductus size. The authors should comment on whether a duct >1.5mm is associated with hemodynamically significant PDA. Do the authors equate a large duct (>1.5mm) to a hemodynamically significant PDA. Hemodynamic compromise in each of the groups should be in Table 1.

We use a large ductus, hence a PDA with a diameter above 1.5 mm to identify a PDA that is known to be associated with morbidity and mortality in extremely preterm neonates (Sellmer et al. ADCF Neonatal 2013). Considerations on the diameter also from 1) M. El Hajjar, G. Vaksmann, T. Rakza, G. Kongolo, L. Storme. Severity of the ductal shunt: a comparison of different markers. Arch. Dis. Child. Fetal Neonatal Ed., 90 (2005), pp. F419-F422 and 2) A. Sehgal, P.J. McNamara Does echocardiography facilitate determination of hemodynamic significance attributable to the ductus arteriosus? Eur. J. Pediatr., 168 (8) (2009), pp. 907-914.

Solely using diameter has increasingly been recognized as a way to identify PDA that are clinical relevant.  

The large European BeNeDuctus trail that compare expectant management with early treatment used diameter of 1.5 mm to define extremely preterm neonates that could be included in the trail (https://trialsjournal.biomedcentral.com/articles/10.1186/s13063-021-05594-x).

Possibly because it is readily obtained on echocardiography. Also, diameter is associated with reversed flow in great arteries (Hsu KH, et al. J Pediatr. 2019) and more advanced echocardiographic markers have not been found to be clinical relevant.

The approach to assessment of the PDA is moving in the direction of not only identifying PDA patency, need of treatment, or quantification of hemodynamic effects but also to identify neonates at increased risk of PDA related morbidity and mortality. We would therefore prefer not to use the term “hemodynamically significant”.

In order to further characterize the PDAs we have included more echocardiographic data in Table 1. And we have updated the Methods section accordingly.

As this is a study and not a review, please delete Figure 1. Normally it is not a good practice to have figures in the introduction unless it is a review. The figure is not adding anything to the manuscript. Also, the reviewer does not see any reference to Figure 1 in the introduction.

We have removed Figure 1 from the manuscript.

Figure 1 – The figure has three parts. 1A – is it the diagram of the fetus or the newborn? 1B – the normal heart. The arrow from 1A to 1B means that the 1B should demonstrate a PDA (with a right to left shunt, if 1A is a fetus) or left to right if 1A is a neonate. Additionally, the figure legend does not explain the three figures in Figure 1. Please delete Figure 1.

We have removed Figure 1 from the manuscript.

Methods –

In the clinical information section, how many babies in the study received caffeine within the first three days of life or prior to sample collection for analysis? Please clarify caffeine use in the NICUs. Caffeine could impact protein biomarkers in the blood. We need information in Table 1.
Caffeine is second to antibiotics the most used drug in the neonatal ward (Stark et al 2022). Caffeine is routinely used for all neonates born at a GA < 32 weeks at the NICU at Aarhus University Hospital. Caffeine is possible involved also in inflammatory response and we find it very relevant to include data on the use of caffeine.

We have included data on use of caffeine in Table 1.

The authors should present more information on the clinical course until discharge. The real question is whether protein biomarkers predict morbidities and mortality at 36 weeks of PMA.
That is a very interesting question. However, the aim of this study was to investigate any differences between proteome in neonates with and without a PDA. Controversies on the PDA has been going on for decades now. Much focus has been on echocardiographic markers and cardiovascular biomarkers. We find that it is important to use a new method – proteomics – in order to demonstrate that the PDA may be more complicated than what is seen on echocardiography.

Closing the PDA does not improve outcome for the extremely preterm neonates. Rather than discussing how and when to close the PDA we think that we need to see the PDA as one of the complications to preterm birth. Our theory is that an open PDA is an indication that this neonates is challenged not only by hemodynamic compromise but possibly also by other important physiological process that are not on track. Proteomics way help to point in the directions to investigate further.

It would be of great interest to evaluate other complications to preterm birth with proteomics.  Possibly several of the complications (IVH, BPD) have a common pathophysiological background – not solely restricted to the cardiovascular system. Could this be inflammation? However, that is not the scoop of this paper.

Line 41 We have revised the introduction to focus on no other complications than the PDA to make it clear that this paper is solely on that complication to extremely preterm birth. And we have made extensive revisions in the Discussion.

How many of the PDAs closed spontaneously in all the groups. Did any infants enrolled in the study treated medically (medications) or by surgery (ligation)? How do they relate to the proteomic change on Day 3?

Blood samples were taken and echocardiography was performed before any of the neonates received any treatment (medically or surgically).

Clinical guideline at study time recommended that all neonates born at a GA < 28 weeks that had a PDA were treated with Ibuprofen (Pedea) irrespective of clinical status, PDA size, or other echocardiographic measures. Unless contraindications for Ibuprofen was present.

In the group of neonates with a large PDA only 4 neonates closed their PDA from day 3 to day 6. We found this number to be too low to do further analysis in relation to findings on proteomics.

The authors should present data on BPD at 36 weeks GA in all the groups. Additionally, other major morbidities such as IVH, NEC, and ROP status at 36 weeks should be presented in the Table.

In Table 1 we only present characteristics present before blood samples were taken and echocardiography was performed day 3.

One could argue that IVH must have occurred before day 3. And we now present data on that. However, NEC, ROP and BPD would not be diagnosed until after day 3 when blood samples and echocardiography was performed.

Please find the revised version of Table 1 now also including data on IVH.

The ventilation status of infants in all the groups on the day of collection of samples is of additional interest. Mechanical ventilation by itself can induce inflammatory change in premature infants.
Use of mechanical ventilation is very relevant and it is clearly a mistake that it was not included.

Table 1 has been revised to also include this information.

Discussion – The reviewers have the following questions regarding lower protein fold differences with small PDA vs no PDA, compared to large PDA vs no PDA –

1.       The infants in the large PDA group were of lower gestational age (Table 1). As far as proteomics is concerned, everyday change in GA will produce a change in plasma proteomics in premature infants. Could the proteomic change be a surrogate marker for gestational age rather than PDA itself?

Ontogenic changes have also been a major concern of ours in the project. However, we have made great efforts in order to evaluate this matter.

1) To take the ontogenic effects into account we have identified PDA-associated protein changes using linear mixed effect models, where the null model was calculated with GA thereby allowing the model to take this effect into account.

2) We re-did all calculations using only samples from the GA 27 group (supplementary figure S3), and found similar results. 3) We have plotted the concentration of all significant proteins vs GA for PDA and non-PDA, respectively. These allows for assessing the effect of GA and ontogeny. We have manually inspected the plots and made them available to the reader.

We have pointed this out in Methods section (Line 165) and mentioned it in Limitations in the Discussion (Line 486).

It would be of great value to repeat these findings in a new larger cohort were the effects of GA could be investigated even more closely.

  1. The authors have to highlight how they made the association between proteomic change and PDA status.
    We have added a comment in the Results section (Line 226) referring to the explanations in the Methods section on statistical analysis.

In the Methods section we explain the statistical analysis used to evaluate the association between PDA status and proteomics (Line 165).

We would be happy to elaborate further if Reviewer 1 find it necessary.

  1. Could PRBC transfusions have contributed to a proteomic change, as more babies in the large PDA group had PRBC transfusions.

We have been able to find proteomics data on PRBC changes over time. But we have not been able to find data on changes in proteomics in patients receiving transfusions. However, it is well known that not only number of red cells increase but also changes in electrolytes and LDH.

It is reasonably to point out in limitations that this is a concern.

Line 486. We have made substantial revisions to the section on Limitations.

4.       Sepsis was higher in the small PDA group; Preeclampsia which could contribute to inflammation was higher in the no PDA group. How do the authors relate the findings to the data in Table 1?

Preeclampsia has traditionally been regarded as a risk factor for PDA, however a meta-analysis from 2021 by Chang Liu et al. found that there was no association between preeclampsia and PDA (https://www.frontiersin.org/articles/10.3389/fped.2020.605879/full).

Sepsis is associated with PDA – also in the described meta-analysis. We found 6 neonates with small PDA had sepsis and 4 neonates with a large PDA and also 4 with no PDA. These are very small numbers. But the main thing is that we did not find that sepsis in neonates with large PDA could explaine the immune response.

We would be cautious to make any conclusions based on this. However, both preeclampsia and sepsis are inflammatory states that could influence the development of the PDA and both have great impact on the neonate and the risk of other important complications to preterm birth.

Line 496. As mentioned already has substantial revisions been made in the section on Limitations.

5.       Inflammatory markers were increased in the large PDA groups, the clinical data presented in Table 1 does not support the finding.
In neonates with large PDA compared to neonates with no PDA we find increased levels of proteins related to inflammation. We find pro- and anti-inflammatory cytokines and we find proteins involved in inflammation or regulated by inflammation that also have other primary tasks (eg. angiotensinogen).

We argue that possibly there is an imbalance that create an inflammatory response that is different in neonates that have a large PDA compared to neonates with no PDA.

We are not convinced that this is due to one single event (sepsis or preeclampsia) but rather the results of multiple factors or a not fully understood common pathophysiological component that makes the difference between neonates that suffer severe long-term complications to preterm birth compared to a more subtle set of complications.

We have made extensive revisions to the Conclusion and the section on Discussion in general in order to make it more clear.

The authors have to expand on the limitations of the study. The study has a lot of limitations as mentioned above.  
We acknowledge that we should elaborate on the limitations of this study. See previous answers.

In the absence of clinical factors related to an infection at birth (such as chorioamnionitis), how do the authors explain changes in inflammatory markers as they relate to PDA?

We hope the response to question 5 also answer this question.

Despite the limitations of the study, the study demonstrates that ductal patency may be influenced by numerous factors. However, is the proteomic change leading to PDA or large PDA leading to proteomic change? The authors should elaborate on this in the discussion, which would be interesting. Also, making the findings clinically relevant will be of additional interest. Presenting more clinical information will help the readers in making reasonable conclusions about this excellent study.

This study was designed to investigate if there are any differences in the proteome in neonates with a large PDA compared to neonates with no PDA. Surprisingly we did find differences not only on proteins known to be involved in hemodynamic status (eg. angiotensinogen) but also in inflammatory and complement systems.

To further characterize the cohort we have included further clinical data on hemodynamic status and IVH.

We have made substantial changes to the conclusion. And we now write: Our findings indicate that the PDA may interfere with or be driven by an imbalance in core bio-physiological systemsEmphasizing that we do not know if changes in proteomics lead to the PDA or if the PDA contribute to the observed changes.

We hope, that this study can give way to further studies in order to further characterize the findings and hopefully answer the question on causality.

Conclusions. L482-484. The first sentence that ‘our study discloses….’ This is the summary of the study. This sentence is apt for the beginning of the discussion so that the readers follow the discussion better. The conclusions should include how the authors interpret the results and make conclusions based on the results.

Line 520. We have removed the first sentence. And revised the conclusion focusing on interpretation rather than summarizing the study.

The second statement – The study provides novel insights into molecular insights…… The reviewer does not agree with that statement. Please delete the statement. The study demonstrates an association between proteomics and PDA status. Association is not causation.

We a very much aware that this study does not in any way demonstrate causation. We have now made further efforts to make sure that we do not give the impression that this study can be used to demonstrate causation.

Line 522. We have removed this line.

Reviewer 2 Report

The authors describe proteomic analyses in a series of extreme preterm infants with and without PDA. The study identified differences in 21 proteins and 8 cytokines in infants with large PDA compared to those without PDA. The authors conclude that the findings associate PDA with alterations in the immune, complement and renin-angiotensin-aldosterone systems.

The manuscript is very detailed and well written. The conclusions reflect the data presented in the results, and add value to existing literature. Please see minor comments below. 

1.  Introduction includes pertinent information. However, there is room to condense by decreasing general information that may not be necessary in this paper. Figure showing PDA seems unnecessary. 

2. Line 62 - Typo: dependant, not 'dependt'

3. Lines 63-64: describing the field of proteomics may not be necessary 

4. Methodology is very detailed and includes pertinent information. Can the authors please clarify the gestational ages of the infants included in the study? The study mentions 28 weeks (line 73) and 32 weeks (line 74). The results include infants <28 weeks. Do the authors intend to suggest the original data source? This can be clarified better. 

5. Line 88 - please clarify here if any other medication use was included or excluded. 

6. Sample collection, preparation and mass spec analysis are described in good detail. 

7. Results - Table 1: Do you mean PRBC transfusion?

8. Line 216 - Statistical filtering criteria - Are the authors referring to the filtering criteria described in the previous section?

9. Lines 267-272: this paragraph seems better suited for discussion than results. 

Author Response

The authors describe proteomic analyses in a series of extreme preterm infants with and without PDA. The study identified differences in 21 proteins and 8 cytokines in infants with large PDA compared to those without PDA. The authors conclude that the findings associate PDA with alterations in the immune, complement and renin-angiotensin-aldosterone systems.

The manuscript is very detailed and well written. The conclusions reflect the data presented in the results, and add value to existing literature. Please see minor comments below. 

We thank reviewer 2 for the comments on the manuscript. We have made changes to the manuscript as detailed in the following.

  1. Introduction includes pertinent information. However, there is room to condense by decreasing general information that may not be necessary in this paper. Figure showing PDA seems unnecessary. 

Line 42 and 66. We have also condensed the general information on both PDA and proteomics as requested.

Figure 1. We realize now that we do not need a figure of the PDA as it is probably known to most readers of this paper. And it has been removed.

  1. Line 62 - Typo: dependant, not 'dependt'

Thank you. This has now been corrected.

  1. Lines 63-64: describing the field of proteomics may not be necessary 

Line 66. We have removed the description of proteomics.  

  1. Methodology is very detailed and includes pertinent information. Can the authors please clarify the gestational ages of the infants included in the study? The study mentions 28 weeks (line 73) and 32 weeks (line 74). The results include infants <28 weeks. Do the authors intend to suggest the original data source? This can be clarified better. 

Line 79. We have rephrased the description of the study group, hopefully making it more clear.

  1. Line 88 - please clarify here if any other medication use was included or excluded. 

Line 95. We have added that also no other medicine was used in order to try to close the PDA.

  1. Sample collection, preparation and mass spec analysis are described in good detail. 

Thank you for this comment also!

  1. Results - Table 1: Do you mean PRBC transfusion?

Yes we do. Sorry. This has now been corrected.

  1. Line 216 - Statistical filtering criteria - Are the authors referring to the filtering criteria described in the previous section?

Line 226: We are referring to the filtering criteria outlines in the Methods section. We have included a comment on this in the manuscript.

  1. Lines 267-272: this paragraph seems better suited for discussion than results. 

We agree. And we have removed the part that is discussion and not results.

Reviewer 3 Report

Sellmer and colleagues report on a study which explores the associations among various proteomic profiles and early postnatal manifestation of patent ductus arteriosus in a cohort of preterm neonates. The study is well planned, the analyses extensive, and the discussion well organized. The redundancy in the methods used increases the credibility of the results.

The methods and results sections are justifiably long in this complex study, but the manuscript could nevertheless be shortened without loss of significant contents, and that might be more appealing to readers.  For example, Figure 1 is an illustration of extremely basic concepts that are well understood by any pediatrician or scientist dealing with perinatal research, and it could easily be deleted. Similarly, the Discussion section presents some excessive details, not immediately relevant to this study.

The Abstract accurately summarizes the contents of the study, and the graphical abstract helps to distill the essential findings, although it cannot be interpreted meaningfully without resorting to the text abstract.

Aside from the superfluous figure, the Introduction is clear and succinct.

The Methods are described in appropriate detail. However, this reviewer would have limited ability to detect technical problems in the proteomic analyses.  

Sepsis is a potentially important confounder, and in the context of this study the authors would likely be focusing on early onset neonatal sepsis; the incidence of this characteristic appears to be surprisingly high in this cohort, and therefore it would be useful if the definition of sepsis was included.  Also, chorioamnionitis is a potential confounder that is related to both fetal and neonatal infection and inflammation, but it is not discussed in the paper.  Do the authors have data on the incidence of clinical and or histological chorioamnionitis?

It is not clear that the relatively narrow range of gestational ages (24 – 27 weeks) significantly contributes to differences in the proteomics profiles observed, but the authors adjusted for this possibility by including GA in the null linear model.

The Results are extensively detailed, and the authors’ decisions to include information in the main paper versus supplementary resources seems appropriate to facilitate reading of the paper.

The Discussion is well organized, but generally verbose, and readers might appreciate more succinct paragraphs. The limitations should include the lack of information regarding chorioamnionitis, if it is unavailable.

Minor issues:

The English language usage is clear, and only a few corrections in grammar, spelling or punctuation will be necessary. Most may be picked up with a careful final reading.  For example:

-          line 42 should read "emphasis" instead of emphasize;

-          line 62, “depends”;

-          line 153, “Solely for the purpose”;

-          line 190, “groups”;

-          Table 1, “PRBC”;

-          Line 231, “Known”;

-          line 326, “increases”;

-          line 327, “through… ???”, sentence incomplete; did the authors mean the endothelium?

-          Line 478, “providing”

Finally, in the title, the semi-colon should be appropriately replaced as a colon.

Author Response

Sellmer and colleagues report on a study which explores the associations among various proteomic profiles and early postnatal manifestation of patent ductus arteriosus in a cohort of preterm neonates. The study is well planned, the analyses extensive, and the discussion well organized. The redundancy in the methods used increases the credibility of the results.

Thank you very much! We appreciate your evaluation and the suggestions made in order to improve the manuscript.

The methods and results sections are justifiably long in this complex study, but the manuscript could nevertheless be shortened without loss of significant contents, and that might be more appealing to readers.  For example, Figure 1 is an illustration of extremely basic concepts that are well understood by any pediatrician or scientist dealing with perinatal research, and it could easily be deleted. Similarly, the Discussion section presents some excessive details, not immediately relevant to this study.

We appreciate your evaluation.

We have focused on removing basic concepts that are well known including Figure 1 and information on PDA and proteomics in the Introduction. We have made great effort to remove details that a not immediately relevant to this study.

The Abstract accurately summarizes the contents of the study, and the graphical abstract helps to distill the essential findings, although it cannot be interpreted meaningfully without resorting to the text abstract.

We have included a new version of the graphical abstract, this will hopefully contribute with information on its own.

Aside from the superfluous figure, the Introduction is clear and succinct.

Thanks! We have made a few revisions based on the comments from the two other reviewers.

The Methods are described in appropriate detail. However, this reviewer would have limited ability to detect technical problems in the proteomic analyses. 

We have made some changes according to comments from reviewer 1 and 2.

Sepsis is a potentially important confounder, and in the context of this study the authors would likely be focusing on early onset neonatal sepsis; the incidence of this characteristic appears to be surprisingly high in this cohort, and therefore it would be useful if the definition of sepsis was included.  Also, chorioamnionitis is a potential confounder that is related to both fetal and neonatal infection and inflammation, but it is not discussed in the paper.  Do the authors have data on the incidence of clinical and or histological chorioamnionitis?

The definition of sepsis is stated in the Methods section line 101 “early onset sepsis defined as 7 days of antibiotics initiated before day 3”. We have added this information to Table 1 to make sure that it is clear.

Line 498. We have included a comment on chorioamnionitis as a possible confounder in the Limitations section.

Unfortunately we do not have data on incidence of clinical and/ or histological chorioamnionitis.

It is not clear that the relatively narrow range of gestational ages (24 – 27 weeks) significantly contributes to differences in the proteomics profiles observed, but the authors adjusted for this possibility by including GA in the null linear model.

Yes, we were concerned that GA may impact the proteome and made effort to handle this.

The Results are extensively detailed, and the authors’ decisions to include information in the main paper versus supplementary resources seems appropriate to facilitate reading of the paper.

In coherence with this, we did not move Results between the main manuscript and the supplementary.

The Discussion is well organized, but generally verbose, and readers might appreciate more succinct paragraphs.

Yes, we acknowledge that more succinct paragraphs improve the readability. We have been through the Discussion section in order to make improvements and to remove details that are not immediately relevant to this study.

The limitations should include the lack of information regarding chorioamnionitis, if it is unavailable.

Line 501.We have included a statement on the lack of data on this in the section on Limitations.

Minor issues:

The English language usage is clear, and only a few corrections in grammar, spelling or punctuation will be necessary. Most may be picked up with a careful final reading.  For example:

We appreciate the errors that are pointed out. They have now all been corrected.

-          line 42 should read "emphasis" instead of emphasize;

check

-          line 62, “depends”;

check

-          line 153, “Solely for the purpose”;

check

-          line 190, “groups”;

check

-          Table 1, “PRBC”;

check

-          Line 231, “Known”;

check

-          line 326, “increases”;

check

-          line 327, “through… ???”, sentence incomplete; did the authors mean the endothelium?

check

-          Line 478, “providing”

check

Finally, in the title, the semi-colon should be appropriately replaced as a colon.

check

Round 2

Reviewer 1 Report

Authors have addressed all the reviewer's concerns.